# Non-Structural Protein 2B of Human Rhinovirus 16 Activates Both PERK and ATF6 Rather Than IRE1 to Trigger ER Stress

**DOI:** 10.3390/v11020133

**Published:** 2019-02-01

**Authors:** Juan Song, Miaomiao Chi, Xiaonuan Luo, Qinqin Song, Dong Xia, Bingtian Shi, Jun Han

**Affiliations:** State Key Laboratory for Infectious Disease Prevention and Control, Collaborative Innovation Center for Diagnosis and Treatment of Infectious Diseases, National Institute for Viral Disease Control and Prevention, Chinese Center for Disease Control and Prevention, 155 Changbai Road, Beijing 102206, China; helen831020@126.com (J.S.); chimiao_1989@126.com (M.C.); xiaonuanluo@126.com (X.L.); sanban1605@163.com (Q.S.); xdgoforit@126.com (D.X.); sbt200888@163.com (B.S.)

**Keywords:** HRV16 2B protein, endoplasmic reticulum stress, PERK, ATF6, IRE1

## Abstract

To understand the underlying mechanisms of endoplasmic reticulum (ER) stress caused by human rhinovirus (HRV) 16 and non-structural transmembrane protein 2B, the expressions of ER chaperone glucose-regulated protein 78 (GRP78) and three signal transduction pathways, including protein kinase RNA-like ER kinase (PERK), activating transcription factor 6 (ATF6) and inositol-requiring enzyme 1 (IRE1), were evaluated after HRV16 infection and 2B gene transfection. Our results showed that both HRV16 infection and 2B gene transfection increased the expression of ER chaperone GRP78, and induced phosphorylation of PERK and cleavage of ATF6 in a time-dependent manner. Our data also revealed that the HRV16 2B protein was localized to the ER membrane. However, both HRV16 infection and HRV16 2B gene transfection did not induce ER stress through the IRE1 pathway. Moreover, our results showed that apoptosis occurred in H1-HeLa cells infected with HRV16 or transfected with 2B gene accompanied with increased expression of CHOP and cleaved caspase-3. Taken together, non-structural protein 2B of HRV16 induced an ER stress response through the PERK and ATF6 pathways rather than the IRE1 pathway.

## 1. Introduction

Human rhinovirus (HRV) infections are major risk factors for asthma and chronic obstructive pulmonary disease (COPD) [1,2,3,4], and asthma affects more than 300 million people worldwide (www.ginasthma.org). The HRV genome is a single-stranded positive-sense RNA that is 7500 nucleotides in length. Its genome encodes a single polyprotein that is proteolytically cleaved to generate various structural and non-structural proteins, including four viral capsid proteins, namely, 1A (viral protein 4, VP4), 1B (VP2), 1C (VP3) and 1D (VP1), and seven non-structural proteins by the viral proteases (2Apro and 3Cpro), namely, 2A, 2B, 2C, 3A, 3B (viral genome-linked protein, VPg), 3C and 3D [5].

Unfolded protein response (UPR) has been documented in plus-strand RNA viral infections [6,7,8,9,10]. For example, hepatitis C virus (HCV) infection up-regulates the activating transcription factor 6 (ATF6) and inositol-requiring enzyme 1 (IRE1)–X-box binding protein 1 (XBP1) pathways [9], and enterovirus 71 (EV71) infection induces the protein kinase R-l like ER kinase (PERK)–eukaryotic initiation factor 2α (eIF2α) pathway. However, XBP1 and PERK activities are repressed in cells that contain HCV replicons, implying that HCV down-regulates these pathways to augment their production of viral proteins [9,11]. West Nile virus induces high levels of XBP1 activation [12]. Therefore, UPR is very complicated for plus-strand RNA viruses.

Multiple physiological and pathological conditions, including hypoxia, redox imbalance, changes in calcium (Ca^2+^) levels and excess protein synthesis, can perturb ER homeostasis, interfere with ER quality control and lead to an accumulation of misfolded and/or unfolded proteins to induce ER stress. Cells cope with ER stress by activation of UPR to maintain ER homeostasis, leading to refolding or degradation of the unfolded/misfolded proteins [13,14,15,16]. There are three signal transduction pathways employing UPR to maintain ER homeostasis. In each pathway, the transmembrane ER stress sensor protein kinase RNA-like ER kinases (PERKs), activating ATF6 and IRE1, are dissociated from BiP (glucose-regulated protein 78, GRP78) to sense abnormal conditions in the ER lumen and transmit the information into the nucleus to up-regulate protein folding machinery and decrease the burden of unfolded proteins on the ER [16,17].

During the viral life cycle, viruses can induce many changes in their host cells, including increased plasma membrane permeability [18], which is mediated by several virally encoded proteins [19,20,21]. Many studies have demonstrated that the 2B protein from enteroviruses facilitates viral release by increasing the concentration of free cytosolic Ca^2+^ from the ER [22,23,24], which is the major intracellular store of Ca^2+^ [25]. Changes in Ca^2+^ levels will perturb ER homeostasis and induce ER stress. As a small hydrophobic protein, the 2B protein of many enteroviruses can form homodimers and homotetramers to localize to the cell membrane system including ER membranes [26,27,28,29]. Based on its localization and proposed potential to modify membrane permeability, we hypothesized that HRV protein 2B might release Ca^2+^ from internal stores to induce ER stress.

To determine whether HRV16 induced ER stress through its non-structural protein 2B, the endogenous expressions of three ER stress sensors, the IRE1, ATF6 and PERK signal pathways were evaluated in this study. We demonstrated that both HRV16 infection and 2B gene transfection could induce ER stress in a time-dependent manner. We further found that both the PERK-eIF2α and ATF6 signaling pathways mediated ER stress rather than the IRE1 pathway. In addition, our results showed that both HRV16 infection and HRV16 2B gene transfection triggered cell apoptosis with increased expression of CHOP and cleaved caspase-3.

## 2. Materials and Methods

### 2.1. Plasmids and Antibodies

The ATF4/6-inducible reporter plasmid (pATF4/6-Luc) was purchased from Addgene (Watertown, MA, USA). Rabbit polyclonal antibodies against GRP78, PERK, p-eIF2a, eIF2a and IRE1 were supplied by Cell Signaling Technology (Danvers, MA, USA). The rabbit monoclonal antibody against cleaved casepase-3 was purchased from Cell Signaling Technology (Danvers, MA, USA). The mouse monoclonal antibody against CHOP was supplied by Cell Signaling Technology (Danvers, MA, USA). The rabbit polyclonal antibody against p-IRE1 was provided by Abcam (Cambridge, England). The rabbit polyclonal antibody against ATF6 was purchased from Santa Cruz Biotechnology (Dallas, TX, USA). Mouse monoclonal antibodies against Flag and β-actin were obtained from Sigma (St Louis, MO, USA). Mouse monoclonal antibodies against HRV VP2 were obtained from QED Bioscience (San Diego, CA, USA).

### 2.2. Computer-Assisted Analysis of Viroporin 2B Sequence

The prediction of TM helices was performed using TMPred (http://www.ch.embnet.org/software/TMPRED_form.html), ∆G Prediction Server (http://dgpred.cbr.su.se/) and SPOCTOPUS (http://octopus.cbr.su.se/).

### 2.3. HRV 16 2B Plasmid Construction

Total RNA was isolated from HRV16-infected H1-HeLa cells 24 h post-inoculation (p.i.). RT-PCR reaction was performed using primers 5′-CGCGGATCCATGGGAATCACTGATTACATACACA-3′ and 5′-CCGCTCGAGTCACTTGTCGTCGTCGTCCTTGTAGTCTTCTTTGTGTATATAAGTTAATTG-3′ harboring *Bam*HI and *Xho*I restriction sites. PCR products were cloned into the pcDNA3.1(+) vector to construct the p2B plasmid with Flag-tag in the C-terminal.

### 2.4. Immunofluorescence Assay

H1-HeLa cells were transfected with 2 μg plasmids using X-tremeGENE™ HP DNA Transfection Reagent (Roche, Basel, Switzerland). At 36 h after transfection, ER was stained by ER-Tracker Red (Invitrogen, Carlsbad, CA, USA) and then incubated accordingly with primary antibodies against Flag. Goat anti-mouse IgG H&L (DyLight^®^ 488) was used as secondary antibodies, and the cells were then stained by DAPI (Merck, Darmstadt, Germany). The slides were visualized using an Olympus FV1000 confocal microscope (Tokyo, Japan).

### 2.5. Virus, Cell Culture and Transfection

The HRV16 strain was propagated at a multiplicity of infection (MOI) of 5 in H1-HeLa cells with 2% serum. H1-HeLa cells were maintained in Dulbecco’s Modified Eagle Medium (DMEM) supplemented with 10% fetal calf serum (FCS). After the cells were seeded 3 h prior to transfection and transfected using X-tremeGENE™ HP DNA Transfection Reagent to make sure there would be approximately 70% transfection efficiency, the cells were harvested and analyzed 12 h, 24 h and 36 h post-transfection. As a spliced XBP1 control test, the cells were cultured in the presence of tunicamycin (1.0 μg/mL) in DMEM containing 2% or 10% fetal bovine serum.

### 2.6. Western Blotting

Total cell extracts were prepared by incubating cells in lysis buffer (Beyontime, Sanghai, China) containing Protease Inhibitor Cocktail Set III (Merck, Darmstadt, Germany) and PhosSTOP EASYpack (Roche, Basel, Switzerland). The protein samples were separated by SDS-PAGE and transferred onto nitrocellulose membranes. The membranes were incubated with primary antibodies against VP2, Flag, GRP78, PERK, IRE1, p-IRE1, eIF2α, p-eIF2α, ATF6, CHOP, cleaved caspase-3 and β-actin. HRP-conjugated goat anti-rabbit/mouse IgG was used as secondary antibodies. Immunoreactive bands were visualized utilizing the Enhanced Chemiluminescence Detection Kit (PerkinElmer, Waltham, MA, USA).

### 2.7. RT-PCR Analysis of XBP1

After total RNA was isolated from cultured cells using TRI Reagent (Sigma-Aldrich, St Louis, MO, USA), RT-PCR was carried out using the SuperScript One Step RT-PCR (Invitrogen, Carlsbad, CA, USA) following the manufacturer’s instructions. The forward primer 5′-CCTTGTAGTTGAGAACCAGG-3′ and reverse primer 5′-GGGGCTTGGTATATATGTGG-3′ were used to examine the expression of XBP1 at the mRNA level. The PCR products were resolved on a 2% agarose gel for electrophoresis.

### 2.8. Luciferase Assay

Approximately 1.5 × 10^4^ cells were plated onto a 96-well cell culture plate 12 h prior to transfection. To examine the effects of HRV16 2B on ER stress, H1-HeLa cells, transfected with pATF6-Luc (0.1 μg/well) or pATF4-Luc (0.1 μg/well), were infected with HRV16 or co-transfected with p2B or pcDNA3.1 (0.1 μg/well). Luciferase activity was measured by the Bright-Glo™ Luciferase Assay System (Promega, Madison, WI, USA) according to the manufacturer’s instructions.

### 2.9. Annexin V-FITC/PI Staining

Annexin V-FITC/ propidium iodide (PI) staining was used to detect the apoptotic effect of HRV infection or p2B transfection on H1-HeLa cells with the Annexin V-FITC Apoptosis Detection kit (Beyontime, Sanghai, China) according to the manufacturer’s protocol. Briefly, the cells were cultured overnight in 24-well plates, then infected with HRV16 (MOI = 5) or transfected with p2B. The adherent cells were washed twice with PBS and incubated with Annexin V-FITC and PI for 15 minutes to achieve double staining. The cells were observed by fluorescence microscopy.

### 2.10. Statistics

All the statistical analyses were performed using the GraphPad Prism 6 software. The significance of variability among the means of the experimental groups was determined by paired Student’s t-test. All values are expressed as the mean ± standard deviation (SD). The differences were considered statistically significant at *p* < 0.05. 

## 3. Results

### 3.1. HRV16 Infection Up-Regulates GRP78 Expression

In response to stress, sequestration of GRP78 by misfolded proteins leads to its release from UPR effector molecules, resulting in UPR activation [30,31]. The induction of GRP78 has been widely used as a marker for ER stress [32]. To determine whether HRV16 infection induced the activation of ER stress response, H1-HeLa cells were infected with HRV16 at a multiplicity of infection (MOI) of 5, the expression of GRP78 was detected by Western blotting. The results showed that HRV16 infection induced the expression of GRP78 in a time-dependent manner (Figure 1A,B). Thus, the results indicated that the ER stress response was induced by HRV16 infection.

### 3.2. HRV16 Infection Activates the ATF6 Pathway

To better characterize the underlying mechanism of the ER stress response induced by HRV16, we examined the effect of infection on the three ER stress pathways (ATF6, PERK and IRE1). During ER stress, the 90-kDa ATF6 protein, which is dissociated from GRP78, is targeted to the Golgi, where it is proteolytically cleaved and its 50-kDa DNA-binding domain is targeted to the nucleus to activate gene expression [33,34,35]. Thus, we examined the cleavage of ATF6 by Western blotting at 3 h, 6 h and 9 h post-infection (MOI = 5) (Figure 1A). The results showed an increased level of the 50-kDa ATF6 fragment in HRV16-infected cells, indicating that HRV16 infection activated the ATF6 pathway (Figure 1A,B). Moreover, the level of cleaved ATF6 in infected cells increased with the extension of infection time (Figure 1A,B). To further confirm the effects of HRV16 on the ATF6 pathway, HRV16-infected H1-HeLa cells (MOI = 5) were transfected with pATF6-luc, an ATF6 luciferase reporter construct, which is a firefly luciferase reporter that contains five copies of the ATF6 consensus binding site upstream of the luciferase gene. Luciferase reporter activity was detected at 3 h, 6 h and 9 h post-infection. The results found that the luciferase activity was significantly increased at 3 h, 6 h and 9 h post-transfection compared with the control cells (Figure 1C). Collectively, these data demonstrated that the ATF6 pathway was activated in H1-HeLa cells upon HRV16 infection.

### 3.3. HRV16 Infection Activates the PERK Pathway

Dissociation of GRP78 from the N-terminus of PERK initiates dimerization and autophosphorylation [36]. Phosphorylation of PERK can be detected as a mobility shift by Western blotting. Activated PERK can phosphorylate eIF2α (p-eIF2α) at S51 [13,37], leading to three downstream effects. To examine whether PERK affected HRV-induced ER stress, we performed Western blotting analysis on HRV16-infected cells (MOI = 5). Our results showed an obvious shift in migration of the PERK band as well as an increase of p-eIF2α in HRV16-infected cells at 3 h, 6 h and 9 h post-infection (Figure 1A). The levels of p-PERK and p-eIF2α were clearly increased in a time-dependent manner (Figure 1A,B). To further determine the effect of HRV16 infection on p-eIF2α effector, ATF4, HRV16-infected H1-HeLa cells (MOI = 5) were transfected with pATF4-luc, an ATF4 luciferase reporter construct. Luciferase reporter activity was detected at 3 h, 6 h and 9 h post-infection. The results showed that the luciferase activity was significantly increased at 3 h, 6 h and 9 h post-infection compared with the control cells (Figure 1D). These results indicated that HRV16 infection induced significant activation of the PERK- eIF2α-ATF4 pathway.

### 3.4. HRV16 Infection Does Not Induce XBP1 Splicing

In response to ER stress, IRE1 is activated by its homo-oligomerization in the membrane, causing the splicing of a 26-bp intron from an evolutionarily conserved ER stress transcription factor XBP1 [16,17]. We assessed the effect of HRV16 infection (MOI = 5) on IRE1 activity at 3 h, 6 h and 9 h post-infection using specific RT-PCR primers to distinguish between the unspliced (inactive, XBP1u) and spliced (active, XBP1s) forms of the XBP1 transcript. We did not detect any spliced XBP1 transcripts in the infected or uninfected cells (Figure 2A). Even with an extension of the infection time to 15 h, spliced XBP1 transcripts were still not detected (Appendix A). However, spliced XBP1 transcripts were detected in the H1-HeLa cells treated with the UPR inducer, tunicamycin (Figure 2A). Therefore, it appeared that HRV16 infection did not induce the expression of the active XBP1 transcript, indicating that IRE1 was not activated upon HRV16 infection.

### 3.5. HRV16 Infection Inhibits Phosphorylation of IRE1

To further understand why IRE1 endonuclease was not activated during HRV16 viral replication, we analyzed the expressions of phosphorylated-IRE1 (p-IRE1) by Western blotting. The results showed that the expression of p-IRE1 was significantly down-regulated in a time-dependent manner compared with the control cells (Figure 2B,C). However, the expression of IRE1 did not increase at 3 h, 6 h and 9 h post-infection (Figure 2B,C). Even when the infection time was prolonged to 15 h, the expression of p-IRE1 was still lower than the normal cells (Appendix A). The activation of IRE1 by its oligomerization promotes autophosphorylation [38], and IRE1 activation enables both its kinase and endoribonuclease activities to transduce signals [38]. Thus, the decrease of p-IRE1 made it difficult to exert the endoribonuclease activity to cleave a 26-bp intron from the XBP1 transcript during HRV16 infection.

### 3.6. HRV16 Infection Induced Apoptosis

CHOP, as an ER stress marker protein, plays an important role in ER stress-induced apoptosis [39,40]. To confirm whether ER stress was induced by HRV16 trigger apoptosis, we detected the protein expression levels of CHOP in H1-HeLa cells infected with HRV16 (MOI = 5) (Figure 3A). The results showed that CHOP was not expressed in the normal cells or in the H1-HeLa cells at 3 h post-infection. However, the expression of CHOP obviously appeared in infected cells at 6 h and 9 h post-infection (Figure 3A). Caspase-3 plays a central role in its cleaved form during the sequential activation of caspases in the execution-phase of cell apoptosis [41]. To confirm whether caspase-3 was activated, the expressions of cleaved casapse-3 proteins were detected in the above infected cells at each infection time points. Similar to the results of CHOP, activated caspase-3 appeared at 6 h and 9 h post-infection (Figure 3A).

If ER stress caused by pathological conditions or microbial infection is overwhelming, the expression of CHOP rises sharply and apoptosis is activated [42]. To further confirm apoptosis in the infected H1-HeLa cells at 3 h, 6 h and 9 h post-infection, apoptosis was measured by Annexin V/ PI double-staining (Figure 3B). We observed that more and more apoptotic cells appeared in H1-HeLa cells infected with HRV16 with the prolongation of infection time (Figure 3B). At 3 h post-infection, the number of apoptotic cells in infected-HeLa cells increased slightly compared with the normal cells. At 6 h post-infection, the number of early apoptotic cells (Figure 3B, green) increased sharply, and some late apoptotic cells appeared (Figure 3B, red). At 9 h post-infection, the early apoptotic cells (Figure 3B, green) increased continuously, while the late apoptotic cells (Figure 3B, red) rose sharply. These data demonstrate that apoptosis occurred in HRV16-infected H1-HeLa cells.

### 3.7. HRV16 2B Protein Is Co-localized with ER in H1-HeLa Cells

The non-structural protein 2B of HRV16 is composed of 95 aa. Using this aa sequence, we analyzed the topology of the integral membrane protein using available algorithms. Although some variability was observed using different algorithms, the transmembrane-encoding region was similar (Figure 4A). The DAS software predicted two hydrophobic regions in the HRV16 2B protein. The first region (39 VKWMLRIISAMVIIIRNSS 57) was predicted to span the membrane with a partial amphipathic cationic helix. The second predicted region (61 TIIATLTLIGCNGSPWRFL 79) was a transmembrane domain (TMD). Therefore, we hypothesized that the HRV16 2B protein might anchor, span and rearrange ER membranes, leading to permeability changes by altering intracellular Ca^2+^ concentrations and triggering ER stress.

To study the effects of the 2B protein on ER stress, H1-HeLa cells were transfected with plasmids p2B conjugated with Flag-tag (Flag-2B), allowing the detection of the expressed protein. Expression of the HRV16 2B protein was confirmed by Western blotting using anti-Flag antibodies at 24 h post-transfection (Figure 4B). When the cells were transfected with plasmids p2B for 24 h, the 2B protein was clearly observed in the ER (Figure 4C). Most of the 2B protein was distributed in the ER compartment, which was labeled by ER-Tracker Red specifically binding to the sulphonylurea receptor of the ER (Figure 4C). These observations confirmed the localization of the HRV16 2B protein in the ER (Figure 4C).

### 3.8. HRV16 2B Induces the Expression of ER Chaperone GRP78

Before the assessment of ER stress triggered by the HRV16 2B protein, the expression of the HRV16 2B protein was first confirmed by Western blotting using anti-Flag antibodies (Figure 5A). Our results showed that the HRV16 2B protein appeared as early as 24 h post-transfection, and its expression level was gradually increased with the extension of transfection time (Figure 5A,B). Next, GRP78 protein was detected in the above p2B-transfected H1-HeLa cells. The results showed that the expression of GRP78 was clearly increased in a time-dependent manner in p2B-transfected cells compared with control cells (Figure 5A,B), suggesting that the HRV16 2B protein induced ER stress in a similar pattern to HRV16 infection.

### 3.9. HRV16 2B Protein Activates Both the PERK and ATF6 Pathways of ER Stress Response

To investigate the link between the ER stress pathways, we first analyzed the phosphorylation levels of PERK and eIF2α in the p2B-transfected H1-HeLa cells. After p2B transfection, the p-PERK increased from 24 h post-transfection and remained high until 36 h post-transfection (Figure 5A,B). The level of p-eIF2α also consistently increased after p2B transfection in a time-dependent manner (Figure 5A,B). To determine the effects of the HRV16 2B protein on ATF4, H1-HeLa cells were co-transfected with either p2B or a control vector and pATF4-luc. The results showed that luciferase activity from transfected p2B was significantly increased at 24 h and 36 h post-transfection compared with the control vector-transfected cells (Figure 5C).

In the meantime, the activated ATF6 pathway was also evaluated in H1-HeLa cells transfected with p2B. The results showed that the active/cleaved form of ATF6 was clearly detected at 24 h and 36 h post-transfection, while it was not found in the cells transfected with the control vector (Figure 5D,E). To further understand whether ATF6 is activated, we co-transfected H1-HeLa cells with p2B and an ATF6 luciferase reporter construct. After the H1-HeLa cells were transfected with both the reporter construct and p2B or a mock vector and harvested at three time points post-transfection, a significant increase of luciferase activity was observed in the cells transfected with p2B compared with the control vector at 12 h, 24 h and 36 h post-transfection (Figure 5F). These results suggest that the HRV16 2B protein induced ER stress to up-regulate the activation of the PERK–eIF2α–ATF4 and ATF6 pathways.

### 3.10. The IRE1 Pathway Is Inactivated in H1-HeLa Cell Transfected with p2B

To understand whether the HRV16 2B protein induced ER stress through the activation of the IRE1α pathway, we examined the effects of HRV16 2B on endoribonuclease IRE1 by Western blotting. Our results showed that the HRV16 2B protein decreased the expression of p-IRE1 at 12 h, 24 h and 36 h post-transfection (Figure 6A,B), suggesting that the IRE1 pathway was not activated in H1-HeLa cells expressing HRV16 2B protein. When the transfection time was prolonged to 48 h, the expression of p-IRE1 was still lower than the normal cells (Appendix A). Similar to HRV16 infection, spliced XBP1 was also not detected in p2B-transfected cells at 12 h, 24 h, 36 h and 48h post-transfection (Figure 6C and Appendix A). Therefore, the HRV16 2B protein did not induce the activation of IRE1 to splice the XBP1 transcript. The results indicated that IRE1 did not regulate ER stress through the 2B protein during HRV16 infection.

### 3.11. Apoptosis Is Induced in H1-HeLa Cell Transfected with p2B

To understand whether the 2B protein induced apoptosis as HRV16 infection, we examined the expression levels of CHOP and cleaved caspase-3 (Figure 7A) in H1-HeLa cells transfected with p2B or pcDNA3.1. Neither CHOP nor cleaved caspase-3 could be detected in H1-HeLa cells transfected with pcDNA3.1 and p2B at 24 h post-transfection. However, both CHOP and cleaved caspase-3 clearly appeared in H1-HeLa cells transfected with p2B at 36 h and 48 h post-transfection. In the meantime, Annexin V/PI double-staining was used to detect apoptotic cells in H1-HeLa cells transfected with p2B at 24 h, 36 h and 48 h (Figure 7B). Similar to HRV16 infection, apoptosis occurred in the H1-HeLa cells transfected with p2B, and the number of apoptotic cells increased in a time-dependent manner. At 24 h post-transfection, the number of apoptotic cells rose slightly compared with the normal cells. At 36 h post-transfection, the number of early apoptotic cells (Figure 7B. green) went up sharply, and some late apoptotic cells (Figure 7B. red) could be observed. At 48 h post-transfection, the early apoptotic cells (Figure 7B. green) increased continuously, while the late apoptotic cells (Figure 7B. red) increased sharply. However, a few apoptotic cells can be observed in the H1-HeLa cells transfected with pcDNA3.1 as a control.

## 4. Discussion

ER homeostasis is maintained by preventing the accumulation of unfolded or misfolded proteins and/or Ca^2+^ depletion [43,44]. Disruption of ER homeostasis triggers the UPR that is an ER-specific cellular stress response [38,43,44]. ER stress can be induced by many pathogenic stress stimuli, such as hypoxia [45], viral infections, ER-Ca^2+^ depletion, protein glycosylation and ER–Golgi vesicular transport [46]. Many plus-strand RNA viruses are reported to induce UPR during viral infection but they have different mechanisms [6,7,8,9,10,12]. In this study, we confirmed that HRV16 induced ER stress through non-structural protein 2B. Using a computational approach, we revealed that the HRV16 2B protein, which has similar characteristics to the PV and CVB3 2B proteins [47], formats hydrophilic pores by monomer polymerization embedding into the ER and Golgi membrane [48]. We postulated that the HRV16 2B protein might also form hydrophilic pores and induce Ca^2+^ permeability but we only confirmed that the 2B protein was localized to the ER membrane in this study.

ER stress mediates UPR signals mainly through three major pathways: ATF6, PERK and IRE1. Therefore, three major pathways were investigated during HRV16 infection in this study. For the PERK branch of UPR, activated PERK phosphorylates eIF2a, which then modulates protein translation to militate the protein overloading in the ER [38,44,49]. Initiation of protein translation is blocked by phosphorylation of eIF2α, blocking the exchange of GDP bound to eIF2α [38]. The upregulation of p-PERK and p-eIF2α was clearly detected at each time point after HRV16 infection and HRV16 2B transfection. In this study, our results indicated that both HRV16 and the nonstructural 2B protein of HRV16 activated the PERK pathway to phosphorylate eIF2α, but this could be achieved using any of the three other cellular eIF2a kinases, namely, PKR, GCN2, HRI (EIFAK1) [50]. Our results were consistent with data on the PERK pathway activated by EV71, vesicular stomatitis virus (VSV) [10,51] and Japanese encephalitis virus (JEV) [52]. Our results also demonstrated that the activation of the PERK pathway contributed to the activation of the transcription factor, ATF4 (Figure 1D and Figure 5C). The PERK–ATF4–CHOP pathway can induce apoptosis by binding to the death receptor pathway [53]. CHOP-induced apoptosis is relevant to microbial infection, including DNA and RNA viruses that cause ER stress [42]. The same results can be confirmed in H1-HeLa cells infected with HRV16.

Many studies have shown that there are complicated mechanisms in the replication of positive-strand viruses [9,54]. TBEV, JEV and DENV infections trigger the IRE1 pathway of UPR, leading to the expression of spliced XBP1 transcripts [54,55]. However, HCV shows an inhibitory effect on the transactivation of XBP1. Similarly, the specific XBP1 transcript cleavage was not detected in cells infected with HRV16 or transfected with plasmids p2B in this study (Figure 2A and Figure 6C). Since IRE1 is activated by its home-oligomerization following autophosphorylation to splice a 26-bp intron from XBP1 [38], phosphorylation of IRE1 should be respectively detected in H1-HeLa cells after HRV16 infection or p2B transfection. However, the results show that down-regulation of phosphorylated IRE1 was observed in H1-HeLa cells after HRV16 infection or p2B transfection, but the IRE1 expression at the protein level was not decreased (Figure 2B and Figure 6A). Since an activation of the IRE1–XBP1 pathway takes more time [53], we extended the time of virus infection from 9 h to 15 h post-infection; we did the same with p2B transfection, but from 36 h to 48 h (Appendix A). However, the expression of p-IRE1 was still down-regulated by Western blotting and XBP1s were not detected by RT-PCR. Thus, IRE1-XBP1 cannot function during ER stress from H1-HeLa cells infected with HRV16.

Taken together, the non-structural 2B protein of HRV16 induced an ER stress response, which was characterized by the induction of the ATF6 and PERK pathways rather than the IRE1 pathway during HRV16 infection. Further studies are necessary to elucidate how both the ATF6 and PERK pathways are activated, as well as the effect of reduced p-IRE1 levels on cell life or death.

## Figures and Tables

**Figure 1 viruses-11-00133-f001:**
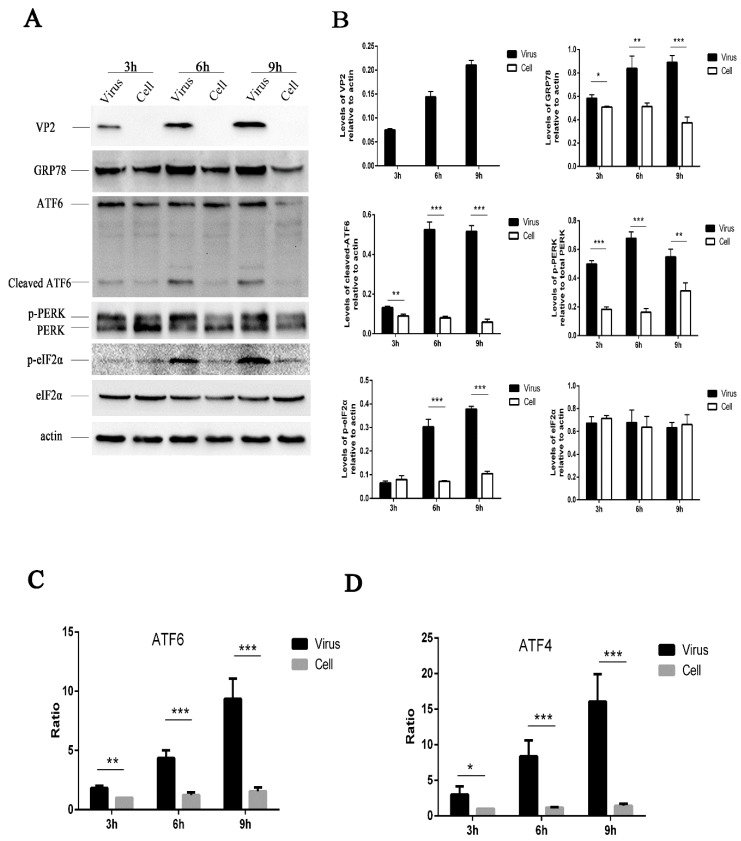
HRV16 infection induces endoplasmic reticulum (ER) stress in H1-HeLa cells. (**A**) After H1-HeLa cells were infected with HRV16 (multiplicity of infection (MOI) = 5) for 3 h, 6 h and 9 h, the expressions of VP2, GRP78, PERK (p-PERK), p-eIF2α, eIF2α, ATF6 (cleavage of ATF6) and actin were detected by Western blotting. Uninfected cells were used as controls. The molecular mass of ATF6 protein and cleaved ATF6 was 90 kDa and 50 kDa, respectively. (**B**) Histogram of gray scanning analyses of the VP2, GRP78, cleaved ATF6, p-eIF2α and eIF2α protein bands of (**A**) relative to actin is shown. The quantitative analysis of the gray value of p-PERK to total PERK was represented. The error bars represent the mean SD of three independent experiments. Statistical differences compared with controls are illustrated as * *p* < 0.05, ** *p* < 0.01, *** *p* < 0.001. (**C**) After the HRV16-infected H1-HeLa cells (MOI = 5) were transiently transfected with ATF6-Luc, the cells were then lysed for the measurement of firefly luciferase activity. Luciferase activities are expressed as the relative value ± SD from three independent experiments. Statistical differences compared with controls are illustrated as * *p* < 0.05, ** *p* < 0.01, *** *p* < 0.001. (**D**) After the HRV16-infected H1-HeLa cells (MOI = 5) were transiently transfected with ATF4-Luc, the cells were then lysed for the measurement of firefly luciferase activity. Luciferase activities are expressed as the relative value ± SD from three independent experiments. Statistical differences compared with the controls are illustrated as * *p* < 0.05, ** *p* < 0.01, *** *p* < 0.001.

**Figure 2 viruses-11-00133-f002:**
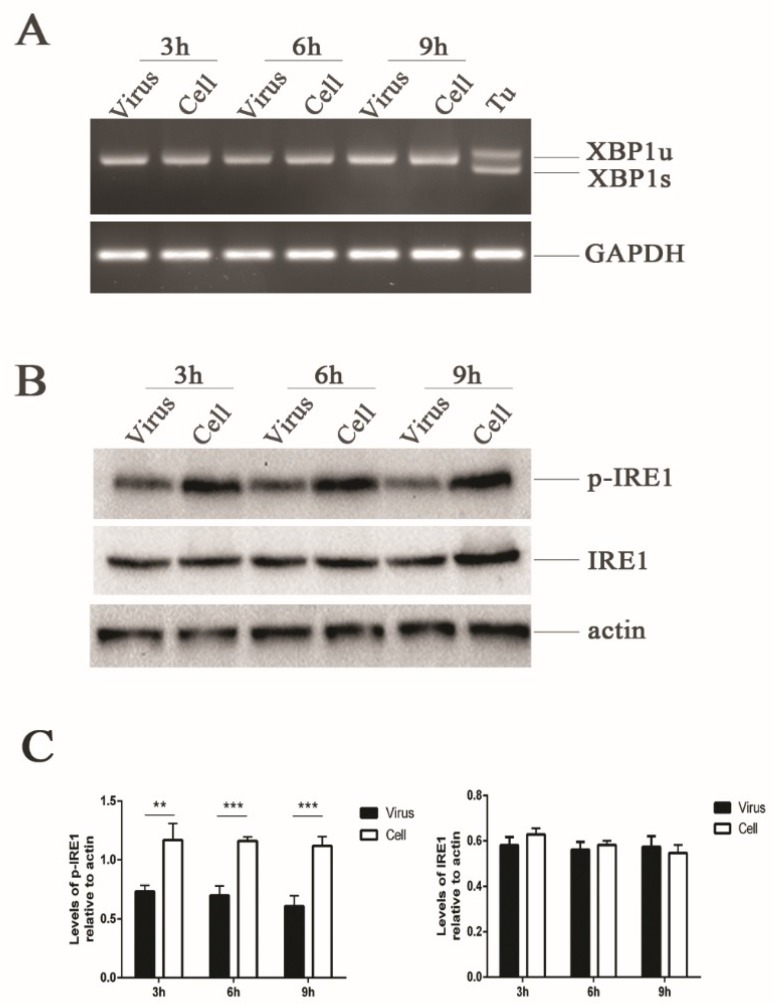
HRV16 infection induces dephosphorylation of IRE1. (**A**) Total RNA was extracted from cells after H1-HeLa cells were infected with HRV16 (MOI = 5) for 3 h, 6 h and 9 h. The expression of XBP1 at the mRNA level was detected by RT-PCR. Cells treated with 1.0 μg/mL tunicamycin were used as a positive control. (**B**) The expressions of p-IRE1, IRE1 and actin were detected in H1-HeLa cells infected with HRV16 (MOI = 5) for 3 h, 6 h and 9 h by Western blotting. Uninfected cells were used as controls. (**C**) Histograms of gray scanning analyses of the p-IRE1 and IRE1 protein bands of (**B**) relative to actin are shown. The error bars represent the mean SD of three independent experiments. Statistical differences compared with the controls are illustrated as * *p* < 0.05, ** *p* < 0.01, and *** *p* < 0.001.

**Figure 3 viruses-11-00133-f003:**
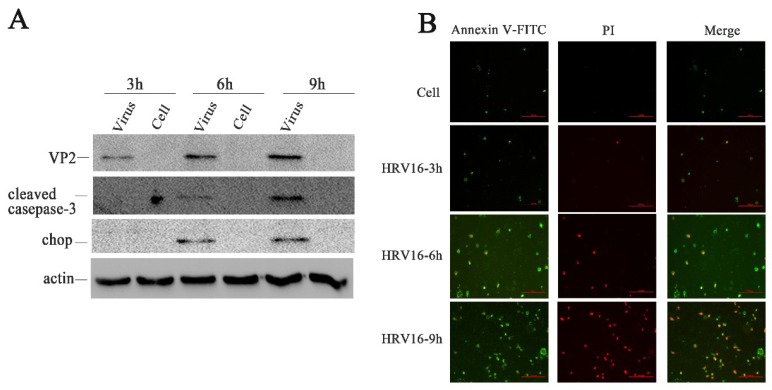
HRV16 infection induced apoptosis. (**A**) The expressions of VP2, cleaved caspase-3, CHOP, and actin were detected in H1-HeLa cells infected with HRV16 for 3 h, 6 h and 9 h by Western blotting. Uninfected cells were used as controls. (**B**) After infection for 3 h, 6 h and 9 h, H1-HeLa cells were stained with FITC-labeled annexin V and propidium iodide. Early apoptotic cells (annexin V+ /PI−) were green. Late apoptotic cells were red and green (annexin V/PI+). Scale bars of images, 200 μm.

**Figure 4 viruses-11-00133-f004:**
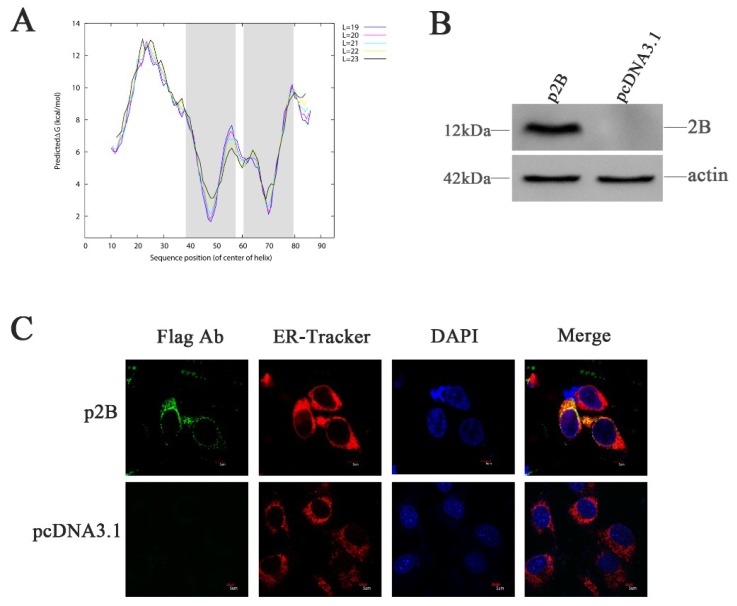
HRV16 2B is localized to the ER. (**A**) Hydrophobic domain analysis of HRV16 2B. The shadow shows the two hydrophobic regions predicted in the protein according to ∆G Prediction Server (http://dgpred.cbr.su.se/). (**B**) H1-HeLa cells were transiently transfected with p2B or pcDNA3.1 plasmid for 24 h. The cells were then lysed and analyzed for the expressions of the 2B protein by Western blotting. (**C**) H1-HeLa cells were transfected with p2B (with Flag-tag). The ER and nuclei were visualized by staining with ER-Tracker Red and DAPI, respectively, and analyzed by a confocal microscope. Scale bars of images, 5 μm.

**Figure 5 viruses-11-00133-f005:**
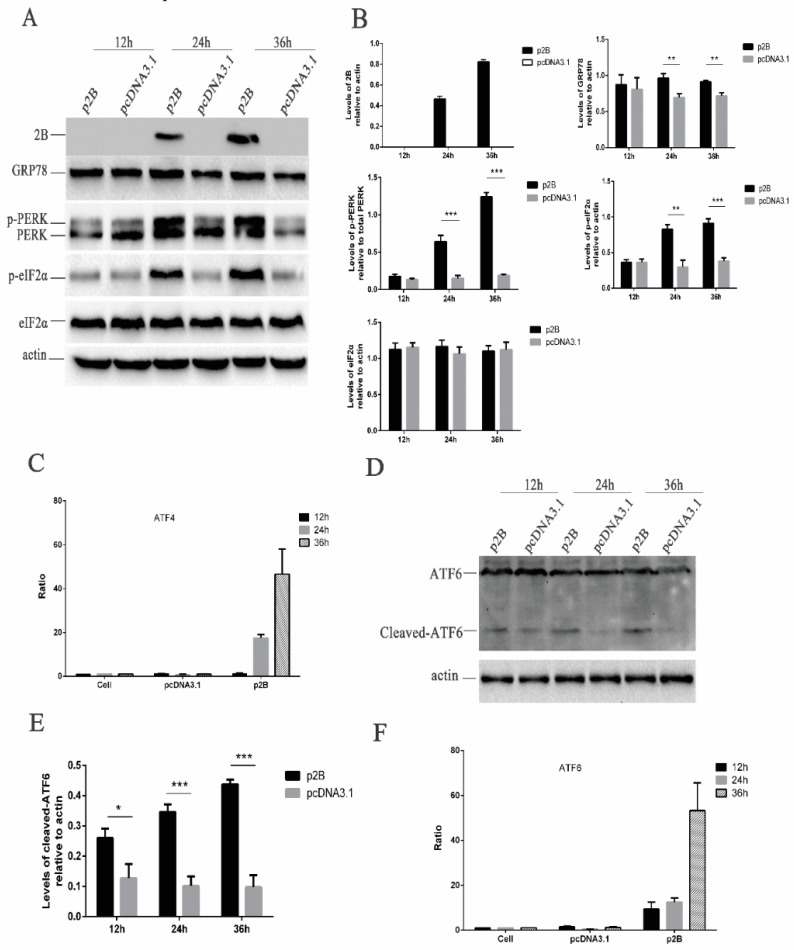
HRV16 2B induces ER stress through the PERK and ATF6 pathways. HRV16 2B activated the PERK–eIF2α–ATF4 and ATF6 pathways. (**A**) H1-HeLa cells were transiently transfected with p2B or pcDNA3.1 plasmid for 12 h, 24 h and 36 h. The cells were then lysed and analyzed for the expressions of the 2B protein, GRP78, PERK (p-PERK), p-eIF2α, eIF2α and actin by Western blotting. (**B**) Histogram of gray scanning analyses of the 2B, GRP78, p-eIF2α and eIF2α protein bands of Figure 4A relative to actin is shown. The quantitative analysis of the gray value of p-PERK to total PERK is represented. The error bars represent the mean SD of three independent experiments. Statistical differences compared with the controls are illustrated as * *p* < 0.05, ** *p* < 0.01, and *** *p* < 0.001. (**C**) H1-HeLa cells were transiently transfected with ATF4-Luc together with p2B or pcDNA3.1. The cells were then lysed for the measurement of firefly luciferase activity. Luciferase activities are expressed as the relative value ± SD from three independent experiments. (**D**) H1-HeLa cells were transiently transfected with p2B or pcDNA3.1. The levels of ATF6 (cleaved ATF6) and actin were detected by Western blotting. (**E**) Histogram of gray scanning analyses of the cleaved ATF6 protein bands of Figure 4D relative to actin is shown. The error bars represent the mean SD of three independent experiments. Statistical differences compared with the controls are illustrated as * *p* < 0.05, ** *p* < 0.01, and *** *p* < 0.001. (**F**) H1-HeLa cells were transiently transfected with ATF6-Luc together with p2B or pcDNA3.1. The cells were then lysed for the measurement of firefly luciferase activity. Luciferase activities are expressed as the relative value ± SD from three independent experiments.

**Figure 6 viruses-11-00133-f006:**
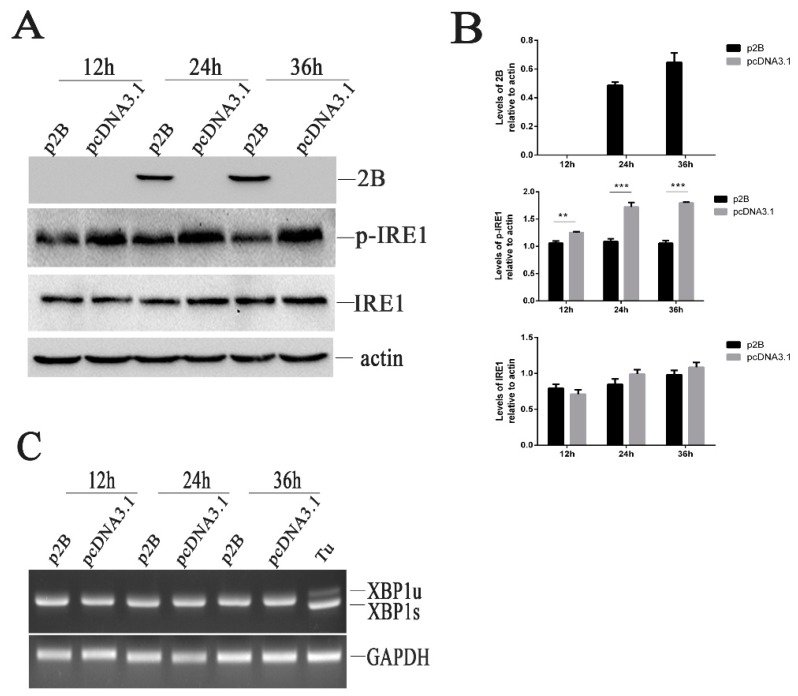
HRV16 2B dephosphorylates p-IRE1 protein. (**A**) H1-HeLa cells were transfected with p2B or pcDNA3.1 for 12 h, 24 h and 36 h. The time course of the 2B, p-IRE1 and IRE1 proteins was detected by Western blotting. β-actin was used as a control. (**B**) Histogram of gray scanning analyses of the 2B, p-IRE1 and IRE1 protein bands of Figure 5A relative to actin is shown, respectively. The error bars represent the mean SD of three independent experiments. Statistical differences compared with controls are illustrated as * *p* < 0.05, ** *p* < 0.01, and *** *p* < 0.001. (**C**) H1-HeLa cells were transfected with p2B or pcDNA3.1 for 12 h, 24 h and 36 h. The time course of XBP1 mRNA expression was detected by RT-PCR. “Tu” represents cells treated with 1.0 μg/mL tunicamycin.

**Figure 7 viruses-11-00133-f007:**
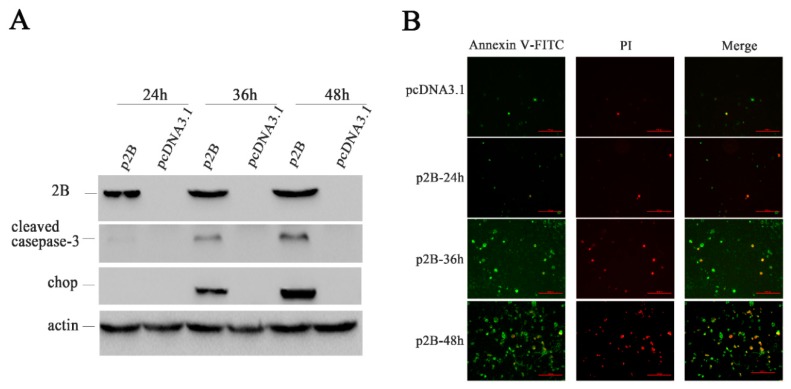
HRV16 2B induced apoptosis. (**A**) The expressions of the 2B protein, cleaved casepase-3, CHOP, and actin were detected in H1-HeLa cells transfected with p2B for 24 h, 36 h and 48 h by Western blotting. Cells transfected with pcDNA3.1 were used as controls. (**B**) After transfection for 24 h, 36 h and 48 h, H1-HeLa cells were stained with FITC-labeled annexin V and PI. Early apoptotic cells are green. Late apoptotic cells are red. Scale bars of images, 200 μm.

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
