# Peer review of "Non-Structural Protein 2B of Human Rhinovirus 16 Activates Both PERK and ATF6 Rather Than IRE1 to Trigger ER Stress"

_viruses, 2019, doi:10.3390/v11020133_

Reviewer 1 Report

Song and colleagues investigated the induction of ER stress pathways in cells infected with human rhinovirus 16 (HRV16). Previous research demonstrated that diverse positive strand RNA viruses, including picornaviruses closely related to HRV16, induce different levels of ER stress activation in different cell/virus systems. The authors demonstrate that infection of HRV16 correlates with a specific pattern of ER stress pathways, which includes cleavage of a transcription factor ATF6 as well as phosphorylation of PERK, but not activation of endonuclease IRE1 and therefore lack of splicing of Xbp mRNA. They also demonstrated that both cleavage of ATF6 and phosphorylation of PERK activates corresponding transcription programs by using luciferase reporter constructs. Interestingly, this pattern was recapitulated upon expression of a viral protein 2B, suggesting that this protein is the major driver of ER stress response in infected cells, although at least two other picornavirus proteins, 2C and 3A (and intermediates containing these sequences) have membrane targeting domains and were previously implicated in alterations of the cellular membrane metabolism. This part is supported by well documented solid data. The second part of the paper however contains experiments lacking important controls, and which are often over interpreted, I recommend removing them from the manuscript (see specific comments below).   

Fig. 1 MOI of infection should be indicated.

Fig. 6 Without a control of mock-infected cells and cell transfected with an empty vector treated with cyclosporine A no conclusion about the specificity of its action in this system can be made. Anyway, the difference in phosphorylation status or IRE1 and its recovery are minimal, I recommend just removing this section.

Fig. 7 This section is not clear. What did the authors mean by “increased expression” of CaN? The amount of the protein? Activity? Chelation of Ca in the cytoplasm is a drastic measure with pleiotropic effects on multiple pathways. Anyway, the specificity of the presented assay is not established and the effect of the chelating agent on the virus infection is not investigated, so it is not clear if the decrease of the measured parameter is related to Ca chelation or inhibition of viral replication. I recommend removing this section.  

Figs. 8 and 9. The same concern as for Fig. 6. There no controls of mock-infected cells or those transfected with an empty vector treated with BAPTA. It is likely that treatment with BAPTA affects ER stess pathways (phosphorylation of IRE-1, PERK and cleavage of ATF-6) regardless of the presence of the viral products. Remove this section.

Other comments:

The text is very repetitive and should be made shorter and clearer. For example, lines 179-186 contain at least four times repeated in various ways “cleavage of ATF6”.

The title should contain either “human rhinovirus 16” or “a human rhinovirus”.

Author Response

Response to Reviewer 1 Comments

Point 1: Fig. 1 MOI of infection should be indicated.

Response 1: MOI has been indicated in the Figure 1.

Point 2: Fig. 6 Without a control of mock-infected cells and cell transfected with an empty vector treated with cyclosporine A no conclusion about the specificity of its action in this system can be made. Anyway, the difference in phosphorylation status or IRE1 and its recovery are minimal, I recommend just removing this section.

Response 2: According to reviewer’s recommendation, we remove this section.

Point 3: Fig. 7 This section is not clear. What did the authors mean by “increased expression” of CaN? The amount of the protein? Activity? Chelation of Ca in the cytoplasm is a drastic measure with pleiotropic effects on multiple pathways. Anyway, the specificity of the presented assay is not established and the effect of the chelating agent on the virus infection is not investigated, so it is not clear if the decrease of the measured parameter is related to Ca chelation or inhibition of viral replication. I recommend removing this section.

Response 3: According to reviewer’s recommendation, we remove this section.

Point 4: Figs. 8 and 9. The same concern as for Fig. 6. There no controls of mock-infected cells or those transfected with an empty vector treated with BAPTA. It is likely that treatment with BAPTA affects ER stess pathways (phosphorylation of IRE-1, PERK and cleavage of ATF-6) regardless of the presence of the viral products. Remove this section.

Response 4: According to reviewer’s recommendation, we remove this section.

Point 5: The text is very repetitive and should be made shorter and clearer. For example, lines 179-186 contain at least four times repeated in various ways “cleavage of ATF6”.

Response 5: We revised the manuscript according to reviewer’s recommendation.

Point 6: The title should contain either “human rhinovirus 16” or “a human rhinovirus”.

Response 6: We revised the title as ” Non-structural protein 2B of human rhinovirus 16 activate both PERK and ATF6 rather than IRE1 to trigger ER stress”

Reviewer 2 Report

The article with title “Non-structural protein 2B of human rhinovirus activate both PERK and ATF6 rather than IRE1 to trigger ER stress by increasing cytoplasmic Ca2+ concentration” by Juan Song et al. demonstrates ER stress caused by rhinovirus infection in vitro using H1-HeLa cells. Non-structural protein 2B plays an important role of triggering ER stress through the PERK and the ATF6 pathways. These activations of UPR responses are mediated by an increase in cytoplasmic Ca2+ concentration. The results are described orderly, and there are few illogical leaps.

Major comments:

This reviewer is interested in molecular mechanisms by which HRV infection and HRV16 2B transfection preferentially phosphorylate PERK rather than IRE1, in spite that the luminal domain of IRE1 is similar to that of PERK. Does calcineurin specifically dephosphorylate IRE1? This question should be addressed in the discussion.

As described in the discussion, the expression of CHOP and apoptosis were not tested in this study. However, it would be relatively easy to evaluate apoptosis by western blotting for apoptotic factors such as CHOP and Caspase-3, or TUNEL assay. This reviewer is interested whether apoptotic pathway is activated by HRV infection and HRV16 2B transfection, because the PERK pathway and ATF4-Luc are significantly activated.

It is well known that an activation of the IRE1-XBP1 pathway takes more time rather than that of the PERK and the ATF6 pathways (Developmental Cell, 2003:4;265). Did authors evaluate the IRE1 pathway by longer duration of HRV infection (12-15hrs) and/or HRV16 2B transfection (48hrs)?

Minor comments:

The concentrations and durations of tunicamycin, and cyclosporin A treatment are not described in the Methods. Also, these reagents were probably added in the culture medium after recovering culture after transfection/infection. Is the duration of recovery 24hrs?

Author Response

Response to Reviewer 2 Comments

Point 1: This reviewer is interested in molecular mechanisms by which HRV infection and HRV16 2B transfection preferentially phosphorylate PERK rather than IRE1, in spite that the luminal domain of IRE1 is similar to that of PERK. Does calcineurin specifically dephosphorylate IRE1? This question should be addressed in the discussion.

Response 1:

Some studies show that calcineurin is a down-stream molecule of PERK signaling by direct interaction with PERK to promote PERK auto-phosphorylation, in the meantime, calcineurin activity is also increased by PERK during ER stress ( PLoS One. 2010; 5(8): e11925. Cellular Signalling  2014;26: 2591–2600. J Biol Chem. 2013; 288(47): 33824–33836.).

Because the other reviewer recommended did not involve in calcineurin in this paper. We removed associated the results of research.

Point 2: As described in the discussion, the expression of CHOP and apoptosis were not tested in this study. However, it would be relatively easy to evaluate apoptosis by western blotting for apoptotic factors such as CHOP and Caspase-3, or TUNEL assay. This reviewer is interested whether apoptotic pathway is activated by HRV infection and HRV16 2B transfection, because the PERK pathway and ATF4-Luc are significantly activated. 

Response 2: Thanks for reviewer’s recommendation. As reviewer’s recommendation, we found that expression of CHOP increased in H1-HeLa cells infected with HRV16 or transfected with p2B plasmid. Cleaved caspase-3 was also found in above H1-HeLa cells (Figure 3 and 7 ). The results showed that apoptotic pathway is activated by HRV infection and HRV16 2B transfection. The results were added in the revised manuscript.

Point 3: It is well known that an activation of the IRE1-XBP1 pathway takes more time rather than that of the PERK and the ATF6 pathways (Developmental Cell, 2003:4;265). Did authors evaluate the IRE1 pathway by longer duration of HRV infection (12-15hrs) and/or HRV16 2B transfection (48hrs)? 

Response 3: According to reviewer’s recommendation, we evaluate the IRE1 pathway in H1-HeLa cells through prolonging infection time to 15h or transfection time for 48hrs. The results showed p-IRE1 did not change significantly compared with control cells, and cleaved XBP1 did not appeared in the any infection time points or transfection time points (Figure S1 and S2). The results can be seen in the revised manuscript.

Point 4: The concentrations and durations of tunicamycin, and cyclosporin A treatment are not described in the Methods. Also, these reagents were probably added in the culture medium after recovering culture after transfection/infection. Is the duration of recovery 24hrs?

Response 4: Tunicamycin was used as positive control for evaluation of spliced XBP1 in this study. The concentrations and durations of tunicamycin treatment was added in the Methods (1.0 μg/mL). According to the other reviewer’s recommendation, both the methods and results associated with cyclosporin A have been removed from the revised manuscript.

Round  2

Reviewer 1 Report

I am satisfied with how the authors addressed my criticism and support publication of the paper.